



# A decade of marine inorganic carbon chemistry observations in the northern Gulf of Alaska – Insights to an environment in transition

Natalie M. Monacci[1], Jessica N. Cross[2], Wiley Evans[3], Jeremy T. Mathis[4], Hongjie Wang[5]

[1]College of Fisheries and Ocean Sciences, University of Alaska Fairbanks, Fairbanks, AK 99775 USA

[2]Pacific Marine Environmental Laboratory, National Oceanic and Atmospheric Administration, Seattle, WA 98815 USA

[3]Hakai Institute, Campbell River, BC V9W 0B7 Canada

[4]Science, Technology and International Affairs Program, Georgetown University, Washington, DC 20057 USA

[5]Graduate School of Oceanography, University of Rhode Island, Narragansett, RI 02882 USA

*Correspondence to*: Natalie M. Monacci (nmonacci@alaska.edu)

**Abstract.** As elsewhere in the global ocean, the Gulf of Alaska is experiencing the rapid onset of ocean acidification (OA) driven by oceanic absorption of anthropogenic emissions of carbon dioxide from the atmosphere. In support of OA research and monitoring, we present here a data product of marine inorganic carbon chemistry parameters measured from seawater samples taken during biannual cruises between 2008 and 2017 in the northern Gulf of Alaska. Samples were collected each May and September over the 10–year period using a conductivity, temperature, depth (CTD) profiler coupled with a Niskin bottle rosette at stations including a long–term hydrographic survey transect known as the Gulf of Alaska (GAK) Line. This dataset includes discrete seawater measurements such as dissolved inorganic carbon and total alkalinity, which allows the calculation of other marine carbon parameters, including carbonate mineral saturation states, carbon dioxide ($CO_2$), and pH. Cumulative daily Bakun upwelling indices illustrate the pattern of downwelling in the northern Gulf of Alaska, with a period of relaxation spanning between the May and September cruises. The observed time and space variability impart challenges for disentangling the OA signal despite this dataset spanning a decade. However, this data product greatly enhances our understanding of seasonal and interannual variability on the marine inorganic carbon system parameters. The product can also aid in the ground truthing of biogeochemical models, refining estimates of sea–air $CO_2$ exchange, and determining appropriate $CO_2$ parameter ranges for experiments targeting potentially vulnerable species. Data are available at https://doi.org/10.25921/x9sg-9b08 (Monacci et al., 2023).



## 1 Introduction

During the last decade, OA has emerged as one of the most prominent topics in marine research and understanding its potential impacts could be vital for the sustainable management of ecosystem services around the world. The global oceans have become progressively acidified (e.g., Feely et al., 2004; Feely et al., 2009; Jiang et al., 2019) over the last two and half centuries as they have taken up approximately one–fourth of the human output of $CO_2$ into Earth's atmosphere (e.g., Sabine and Tanhua, 2010; Friedlingstein et al., 2022). At present, the decrease in the average global surface ocean pH averages ~0.1 units, making surface ocean acidity 30% higher than at the start of the Industrial Revolution (Gruber et al., 2023; Jiang et al., 2019). This decline in ocean pH and the sharp rise in seawater carbon dioxide ($CO_2$) has led to broad reductions in carbonate mineral concentrations in the oceans with diverse, but often detrimental consequences for both pelagic and benthic species (Andrade et al., 2018; Barton et al., 2012; Bechmann et al., 2011; Cooley et al., 2009; Hurst et al., 2012; Hurst et al., 2013; Hurst et al., 2019; Long et al., 2013; Wright–LaGreca et al., 2022).

While acidification is a global phenomenon in the open ocean, there are regional hotspots where natural coastal processes can precondition waters to have lower pH and carbonate mineral saturation states ($\Omega$), creating additive vulnerabilities. Regions where OA is amplified by concurring anthropogenic and natural processes are also interacting with changing temperature, salinity, and gas solubility (Feely et al., 2018). This is particularly true in the high–latitude north Pacific Ocean (e.g., Byrne et al., 2010; Dore et al., 2009) where studies have shown that the region is already experiencing seasonal events where carbonate minerals, such as the biologically important aragonite, can become undersaturated (Evans and Mathis, 2013; Evans et al., 2013; Evans et al., 2014). Models project that under current emission rates large areas of the subarctic Pacific could become undersaturated with respect to aragonite ($\Omega_A < 1$) by as early as mid–century and the entire region will be perennially undersaturated by 2100 (e.g., Gruber et al., 2023; Mathis et al., 2015). The seasonal effects of glacial runoff may also intensify acidification processes because these waters may have naturally low carbonate mineral concentrations depending on how they are discharged into the marine environment (Reisdorph et al., 2014; Evans et al., 2014). Consequently, high–latitude regions provide a real–time laboratory for the evaluation of potential impacts to organisms, and the associated marine resources that they provide (Mathis et al., 2015).

The northern Gulf of Alaska (NGA) is a predominantly downwelling system (Royer and Emery, 1987) with seasonal exceptions. When the onshore surface Ekman transport relaxes in the summer, deep, nutrient–rich, cold water can inundate the continental shelf below the surface layer near shore (Weingartner et al., 2005). The new intermediate water, also enriched in $CO_2$, causes the saturation horizon for both aragonite and calcite minerals to shoal to within 100 m and 250 m from the surface, respectively (Feely et al., 2004). Along the West Coast of North America, intense wind–driven upwelling can cause the saturation horizon to shoal to the surface over the continental shelf, leaving large, inshore areas exposed to conditions that have been shown to be corrosive to some calcifiers (Comeau et al., 2010; Bednaršek et al., 2012; Bednaršek et al.,



2014). The phenomenon was first observed in the California Current System (Feely et al., 2008), but has also been observed in other high–latitude regions (e.g., Bednaršek et al., 2012; Mathis et al., 2012).

The NGA supports a diverse ecosystem that includes some of the largest commercial fisheries in the world and serves as a pathway for biogeochemical preconditioning of the Bering Sea and Pacific–Arctic region (Cross et al., 2018). The Bering
Sea region, like the NGA, supports a highly productive system and these two large marine ecosystems support the largest single–species fishery in the world, walleye pollock (Fissel et al., 2021). On the relatively shallow (<60 m) Bering Sea shelf, the natural coupling between intense primary production during phytoplankton blooms and respiration of exported organic matter in bottom waters leads to conditions where $\Omega_A < 1$ for months at a time (Mathis et al., 2011a; Mathis et al., 2011b; Mathis et al., 2014), and localized retention of $CO_2$ can lead to calcite undersaturation ($\Omega_C < 1$; Cross et al., 2013).

When all these factors are considered, it becomes obvious that the NGA is at a critical intersection point for coastally driven OA processes. This issue is of particular importance for the people that rely on marine resources. Alongside these natural vulnerabilities, anthropogenic perturbations may cause ecosystem–level shifts that have the potential to decrease the economic value of commercial fisheries. More than 60% of the catch by weight of U.S. fisheries are from Alaskan waters (NMFS, 2022) and impacts on commercial and subsistence fishing are equally important topics in marine resources
conversations in Alaska (Frisch et al., 2015; Szymkowiak and Steinkruger, 2023). Many communities, particularly those in the NGA region, are vulnerable to risks associated with OA through the loss of culture, jobs, income, and food security (Mathis et al., 2015). Several economically important species have shown negative effects of OA on various developmental stages (e.g., Long et al., 2013; Hurst et al., 2019). Experiments on the economically critical red king crab fishery led to a population dynamics model to predict potential effects on fishery yield (Punt et al., 2014; Punt et al., 2021) and the economic
impacts to Alaska (Seung et al., 2015).

To better understand the processes that influence inorganic carbon chemistry and in turn, OA in the NGA, we assembled this decadal time–series data product. Ocean biogeochemical observations, like those we present in this dataset, help researchers determine biogeochemical model performance (e.g., Siedlecki et al., 2017; Hauri et al., 2020), calculate sea–air $CO_2$ flux (e.g., Evans et al., 2013; Gruber et al., 2023), and set ranges to determine physiological responses to OA on specific species
(e.g., Hurst et al., 2013; Long et al., 2013). In turn, these applications can aid in understanding cascading societal impacts of changes in the marine carbonate system, such as determining the vulnerability of a region's marine resources or the effectiveness of marine carbon dioxide removal. We will show how mean water column structure and circulation patterns impact variability in carbon chemistry, discuss some long–term changes in carbon chemistry parameters, and assess the drivers contributing to these patterns of variability as examples of how this data product may be used. In general, we find a
pattern of decreasing surface $CO_2$ during the spring season over time. While our data suggest that this temporal shift is unrelated to temperature patterns, additional research will be required to document a hypothesized change in the timing or



magnitude of the spring bloom that could be leading to these changes. In general, we find that this dataset is good for exploring natural environmental variability in the physical system, and the impacts on carbon parameters.

## 2 Methods

### 2.1 Study Area

The GAK Line (Fig. 1) is a long–term oceanographic time–series that has been sampled at least biannually since 1997. The GAK Line is often colloquially referred to as the Seward Line (SL), since the research cruises begin and end from the University of Alaska Fairbanks (UAF) Seward Marine Center in Qutekcak (Seward), Alaska. The SL program began as part of the Northeast Pacific Global Ocean Ecosystem Dynamics (GLOBEC) program, evolved under a funding consortium from 95 2005–2017, and is currently part of the Northern Gulf of Alaska Long Term Ecological Research (NGA–LTER) project. The northernmost station, GAK1, is at the mouth of Resurrection Bay, with major stations spaced ~20 km apart across the continental shelf to GAK15. In addition to visiting the major stations, most cruises in this dataset began at an inner station, RES2.5, located within Resurrection Bay, and occasionally visited the intermediate (i) stations, spaced every ~10 km from GAK1i to GAK9i.

The GAK Line has two dominant westward circulation features: the Alaska Coastal Current (ACC) and the Alaskan Stream (Fig. 1). The ACC hugs the coast and is driven by freshwater discharge (Childers et al., 2005; Reed, et al., 1987; Royer, 1975) and the Alaska Stream (AS) flows along the continental slope and is driven by the Alaska Gyre (Ladd et al., 2016; Stabeno et al., 2004; Weingartner et al. 2005). The Alaska Gyre controls the circulation in the Gulf of Alaska basin, which responds to the Aleutian Low pressure system and the Pacific Decadal Oscillation. Strengthened seasonal downwelling over 105 the shelf (Royer and Emery, 1987) results from cyclonic wind stress over the Gulf of Alaska basin from the Aleutian Low. Stronger winds and storms associated with the Aleutian Low are typical in autumn, winter, and spring. The resulting downwelling restricts the ACC to a narrow, deep band along the coast (Weingartner et al., 2005) and constructs a shelf break front along the AS (Fig. 2a). Downwelling winds subside in the summer, broadening and shoaling the ACC (Fig. 2b) and allowing a subsurface inflow of cold, nutrient rich Pacific water onto the shelf (Childers et al., 2005; Weingartner et al., 110 2005). This seasonal relaxation of downwelling presumably allows for longer on–shelf summer residence time of subsurface water that may experience elevated rates of organic matter respiration fueled by high summertime primary productivity.

The Bakun upwelling indices are calculated based on Ekman's theory of mass transport due to wind stress (Schwing et al., 1996; Bakun 1973, 1975). Positive values of the Bakun upwelling index are, in general, indicative of upwelling, while negative values imply downwelling. Relative to other upwelling indices broadly available, the Bakun index uses sea level 115 pressure fields from an atmospheric reanalysis to derive estimated near–surface winds. Other indices use winds directly from atmospheric reanalysis that assimilate satellite and *in situ* wind measurements, both of which are limited in the Gulf of



**Fig. 1: Map and circulation of study area in the Northern Gulf of Alaska (adapted from Reed et al., 1987). Main Gulf of Alaska**
**(GAK) Line stations 1–15 are mostly sampled each cruise (filled circles), intermediate (i) stations are sampled periodically (open circles). Cruises begin and end in Qutekcak (Seward), AK and occur mostly during the months of May and September. Most cruises also sampled repeat stations in western Prince William Sound (PWS) including the glacially influenced stations in Icy Bay (IB) and near the Columbia Glacier (CG). Nearshore stations are defined as GAK1–2, middle is GAK3–7, and offshore is GAK8– 15.**




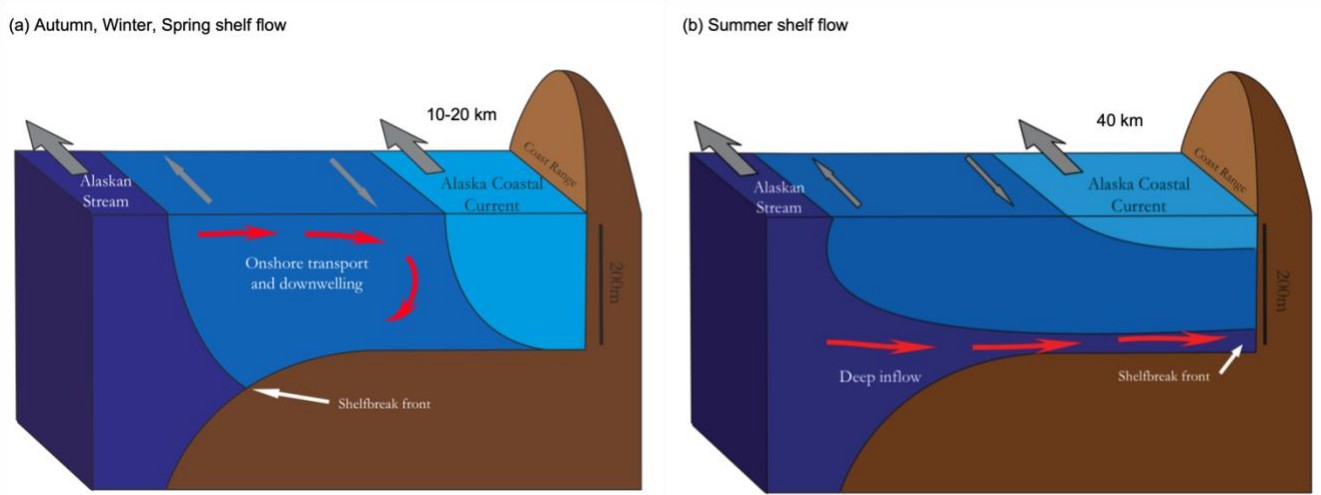

**Fig. 2: Northern Gulf of Alaska shelf flow. a) Autumn, Winter, and Spring shelf flow characterized by onshore transport and downwelling with a relatively narrow (10–20 km) and deep (200 m) Alaska Coastal Current. b) Summer shelf flow characterized by a relaxation of downwelling, allowing the deep inflow of Alaska Stream water under a broader (40 km) and shallower (50 m) Alaska Coastal Current (adapted from Shake 2011).**

Alaska basin. However, there are important limitations to the Bakun product: (1) it does not fully account for upwelling driven by wind stress curl (e.g., upwelling associated with alongshore changes in wind), and (2) it does not account for cross–shore geostrophic transport. Other upwelling indices have been developed for the North American west coast (e.g., CUTI and BEUTI, Jacox, 2018) which resolve some of these challenges, but which are not available for this geographic region. To provide context for this dataset, here we show annual cumulative downwelling by plotting the cumulative sum of the Bakun Index values over each year (Fig. 3). Bakun upwelling indices for 60° N were obtained from the NOAA Environmental Research Division website: https://oceanview.pfeg.noaa.gov/products/upwelling/dnld. The Cumulative Daily Upwelling Indices (CDUI) for our study area and time show the magnitude and duration of downwelling conditions experienced in the NGA through the year with, importantly, a period of relaxation nominally between May and October.

### 2.2 Data Collection and Analysis

The biannual cruises in this dataset were typically completed in May and September from 2008 to 2017. Weather, instrumentation, and personnel were the limiting factors to visiting all sampling stations for each cruise. All data presented here were collected using a Seabird 911 Plus conductivity, temperature, and depth (CTD) profiler with a rosette sampler holding twelve, 5 L Niskin bottles. A Seabird 43 dissolved oxygen sensor was included for all cruises after Fall 2011 (TXF11). Temperature (T, ITS–90), Salinity (S, PSS–78), and Oxygen ($O_2$, Tau and hysteresis corrections applied) bottle data presented here were processed from the upcast profiles using Seabird Scientific Seasoft V2 software.



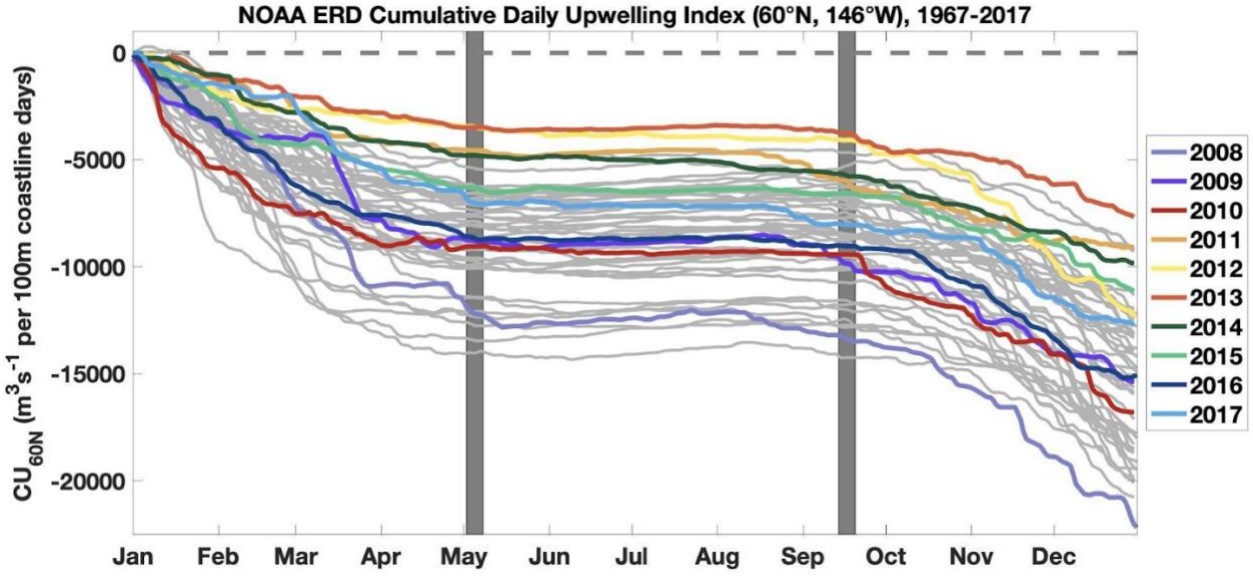

**Fig. 3: National Oceanic and Atmospheric Administration (NOAA) Environmental Research Division (ERD) Cumulative Daily**
**Upwelling Index at 60°N, 146°W from 1967 to 2017 (adapted from https://oceanview.pfeg.noaa.gov/products/upwelling/intro).**
**Negative cumulative (CU) values indicate a downwelling system. Study years (2008–2017, color) are highlighted from the full index**
**years (gray). Average cruise dates in May and September are indicated by shaded, vertical bars.**

Discrete seawater samples were collected from the Niskin bottles to be analyzed in the laboratory for $O_2$, dissolved inorganic
carbon (DIC), total alkalinity (TA), nutrients (nitrate, nitrite, phosphate, and silicic acid), and stable oxygen isotope ratio
($\delta^{18}O$) of seawater. When collected, discrete $O_2$ samples were drawn from the Niskin first, following the GO–SHIP Repeat
Hydrography Manual (Langdon, 2010) protocol, and analyzed using the Winkler method (Carpenter, 1965; Winkler, 1888)
on a Langdon Amperometric titrator at the Ocean Acidification Research Center (OARC) at UAF.

Following the discrete $O_2$ collection, discrete seawater samples for DIC and TA were drawn from the Niskin into rinsed,
borosilicate glass bottles, fixed with saturated mercuric chloride solution, and sealed until analyzed. All DIC and TA samples
were analyzed at the OARC at UAF, but methods vary depending on years. DIC samples collected from 2008 to 2013 were
analyzed by coulometric titration using a Versatile Instrument for the Determination of Total Alkalinity (VINDTA) 3C
paired with a UIC Coulometer. DIC samples collected from 2014–2017 were analyzed using infrared detection with an
Automated InfraRed Inorganic Carbon Analyzer (AIRICA) paired with a LiCOR 7000. TA samples were analyzed in an
open cell by potentiometric titration using a VINDTA 3C or 3S. All DIC and TA measurements were routinely calibrated
using Certified Reference Materials (CRM) from the Dickson Laboratory at the Scripps Institute of Oceanography. Repeat
analyses and comparisons across OARC equipment show there was no analytical difference across the various
instrumentation used, additional discussion on error is included in Sect. 2.3.



Discrete samples for nutrient analyses were collected and analyzed by various laboratories using similar, though distinct

methods. Samples for nutrient analyses collected during cruises in 2008, 2009, 2010, and 2012 were drawn into rinsed, high–density polyethylene (HDPE) vials, frozen at sea, and analyzed at the Whitledge Laboratory at UAF. Here, samples were thawed immediately prior to analyses using colorimetric techniques on a Technicon AutoAnalyzer II and Alpkem model 300 continuous nutrient analyzers (Whitledge et al., 1981; Childers et al., 2005). Samples for nutrient analyses collected during cruises in 2011 and 2013–2015 were syringe–filtered (using 25 mm, cellulose acetate filters with a 0.45 µm pore size) into

HDPE vials, frozen at sea, and analyzed at the National Oceanic and Atmospheric Administration (NOAA) Pacific Marine Environmental Laboratory (PMEL). Here, samples were thawed 3–12 hours before analyses according to the methods of Gordon et al. (1993) using a combination of analytical components from Alpkem, Perstorp, and Technicon. Samples for nutrient analyses collected during cruises in 2016 were also filtered using the previously described method but analyzed at OARC at UAF. Here, samples were thawed overnight before being analyzed according to the methods of Gordon et al.

(1993) using a SEAL Analytical AA3 continuous flow analyzer. Samples for nutrient analyses collected during cruises in 2017 were filtered and analyzed at the Nutrient Analytical Facility (NAF) at UAF. Here, samples were thawed overnight and brought to room temperature before analyses on a Seal Analytical QuAAtro39 continuous flow analyzer according to methods of Armstrong et al. (1967) and Murphy and Riley (1962). The data we report here are from four macronutrient variables: nitrate, nitrite, phosphate, and silicate (heretofore silicic acid). Silicate data from cruises in 2011 and 2013–2017

had a second analysis performed the following day to avoid low values caused by potential polymerization during frozen storage (Burton et al., 1970; Macdonald et al., 1986; Zhang and Ortner, 1998). We believe results from the differing methods used by the various laboratories are mostly comparable, including the filtered vs. non–filtered sampling methods, except for the repeat silicate analysis. Additional discussion of the variability in these methods is included below in Sect. 2.3 and 2.4.

Stable oxygen isotope ($\delta^{18}O$) samples were collected in borosilicate glass vials with no headspace, sealed with parafilm, and

submitted to the Stable Isotope Laboratory (SIL) at Oregon State University (OSU). All $\delta^{18}O$ values are reported relative to Vienna–standard mean ocean water (VSMOW).

**2.3 Data organization and manipulation**

This data product merges the GAK Line data from 20 cruise–level archived datasets (Table 1; Monacci et al., 2020a–j) and is organized using best practice data standards including column header abbreviation standards from Jiang et al. (2022; Table

2). A sample identifier (sample ID) is generated to create a unique value to easily incorporate data into data products such as the Coastal Ocean Data Analysis Product in North America (CODAP–NA; Jiang et al., 2021). The sample ID uses the formula outlined by Jiang et al., 2021 and is calculated following Eq. (1):

$Sample\ ID = Station\ ID\ \times\ 10{,}000\ +\ Cast\ number\ \times\ 100\ +\ Rosette\ position$          (1)





| EXPOCODE | Cruise | Ship | Start Date | End Date | DOI | CRM batch | TA | DIC | Nutrients | O₂ | δ¹⁸O | i |
|---|---|---|---|---|---|---|---|---|---|---|---|---|
| 33A020080502 | TXS08 | M/V Tiglax | 5/2/08 | 5/6/08 | https://doi.org/10.25921/44kh-zb66 | 82 | | | UAF-WL* | | | |
| 33A020080913 | TXF08 | M/V Tiglax | 9/13/08 | 9/16/08 | https://doi.org/10.25921/44kh-zb66 | 82 | | | UAF-WL* | | | |
| 33A020090503 | TXS09 | M/V Tiglax | 5/3/09 | 5/8/09 | https://doi.org/10.25921/n43y-9r47 | 96 | | | UAF-WL* | | | x |
| 33A020090914 | TXF09 | M/V Tiglax | 9/14/09 | 9/19/09 | https://doi.org/10.25921/n43y-9r47 | 96 | | | UAF-WL* | | | x |
| 32QO20100503 | N2S10 | R/V Norseman II | 5/3/10 | 5/7/10 | https://doi.org/10.25921/avxr-m571 | 99 | | | UAF-WL* | | | |
| 33A020100914 | TXF10 | M/V Tiglax | 9/14/10 | 9/18/10 | https://doi.org/10.25921/avxr-m571 | 99 | | | UAF-WL* | | | x |
| 33A020110506 | TXS11 | M/V Tiglax | 5/6/11 | 5/9/11 | https://doi.org/10.25921/07yn-b044 | 108 | | | PMEL | | | |
| 33A020110915 | TXF11 | M/V Tiglax | 9/15/11 | 9/19/11 | https://doi.org/10.25921/07yn-b044 | 112 | | | PMEL | x | x | x |
| 33A020120503 | TXS12 | M/V Tiglax | 5/3/12 | 5/9/12 | https://doi.org/10.25921/mttc-gc63 | 114, 119 | 6.85 | 6.32 | UAF-WL* | x | x | x |
| 33A020120913 | TXF12 | M/V Tiglax | 9/13/12 | 9/18/12 | https://doi.org/10.25921/mttc-gc63 | 120 | 4.3 | 3.9 | UAF-WL* | x | | x |
| 33A020130427 | TXS13 | M/V Tiglax | 4/27/13 | 5/8/13 | https://doi.org/10.25921/n5dy-h455 | 121, 126 | 3.15 | 5.18 | PMEL | x | x | x |
| 33A020130914 | TXF13 | M/V Tiglax | 9/14/13 | 9/26/13 | https://doi.org/10.25921/n5dy-h455 | 128, 129 | 2.68 | 2.15 | PMEL | x | x | |
| 33A020140503 | TXS14 | M/V Tiglax | 5/3/14 | 5/8/14 | https://doi.org/10.25921/ke3w-xp11 | 134, 135 | 4.21 | 2.98 | PMEL | x | x | |
| 33A020140913 | TXF14 | M/V Tiglax | 9/13/14 | 9/19/14 | https://doi.org/10.25921/ke3w-xp11 | 140, 141 | 12.72 | 11.56 | PMEL | x | | |
| 33A020150505 | TXS15 | M/V Tiglax | 5/5/15 | 5/11/15 | https://doi.org/10.25921/r7kp-0j46 | 144 | 8.56 | 7.51 | PMEL | x | | |
| 33A020150914 | TXF15 | M/V Tiglax | 9/14/15 | 9/20/15 | https://doi.org/10.25921/r7kp-0j46 | 144, 146 | | | PMEL | x | | |
| 33A020160430 | TXS16 | M/V Island C† | 4/30/16 | 5/27/16† | https://doi.org/10.25921/ed32-3h29 | 146, 148 | 2.57 | 4.59 | UAF-OARC | x | | |
| 33A020160915 | TXF16 | M/V Tiglax | 9/15/16 | 9/20/16 | https://doi.org/10.25921/ed32-3h29 | 148 | | | UAF-OARC | x | | |
| 33A020170501 | TXS17 | M/V Tiglax | 5/1/17 | 5/8/17 | https://doi.org/10.25921/rvm1-vj65 | 157, 165, 169 | 5.81 | 7.22 | UAF-NAF | x | | |
| 33A020170916 | TXF17 | M/V Tiglax | 9/16/17 | 9/22/17 | https://doi.org/10.25921/rvm1-vj65 | 164, 165, 170 | 4.59 | 4.02 | UAF-NAF | x | | |

**Table 1: List of cruises used to produce this data product. EXPOCODE: Expedition code consists of the four–digit International Council for the Exploration of the Sea (ICES) platform code and the date of departure from port (UTC) in YYYYMMDD. Cruise: The project Cruise name, including the letters of the ship (TX = R/V *TiĝlaX̂*), the northern hemisphere season (S = spring, F = Fall), and the year in format YY (2008 = 08). Ship: Vessel name used for field work, where † indicates the R/V *TiĝlaX̂* experienced engine problems and the M/V *Island C* was chartered in late May to occupy stations GAK2–15, though no DIC/TA samples were collected. Start Date and End Date: UTC in YYY-MM-DD. DOI: Digital Object Identifier (Monacci et al., 2023a-j). CRM Batch: Batch numbers of the Certified Reference Materials. TA and DIC: Mean uncertainty of triplicate discrete samples in µmol kg⁻¹. Nutrients: Laboratory where analysis of discrete nutrient samples were analyzed (UAF-WL = University of Alaska Fairbanks Whitledge Lab, PMEL = Pacific Marine Environmental Laboratory, UAF-OARC = UAF Ocean Acidification Research Center, UAF-NAF = UAF Nutrient Analytical Lab), where * indicates all nutrient values are QF = 3 and are not included in this merged data product. O₂, δ¹⁸O, i: Dissolved oxygen, stable oxygen isotope ratio, and GAK Line i stations, where x indicates data was collected. All files include data with variables CTDTEMP, CTDSAL, DIC†, TA†, Silicate, Phosphate, Nitrate, and Nitrite, except when indicated otherwise. Columns O₂ and δ¹⁸O marked with x indicate that variables CTDOXY, OXYGEN, and DEL18O are included. Column GAK i marked with x indicate the GAK intermediate (i) stations were visited.**

| Abbreviation | Variable description | Unit |
| --- | --- | --- |
| CTDPRS | Pressure recorded from sensors on CTD. | dbar |
| CTDTMP_ITS90 | Temperature on the International Temperature Scale of 1990 (ITS-90) from the sensors on CTD. | °C |
| CTDSAL_PSS78 | Salinity on the Practical Salinity Scale 1978 (PSS-78) calculated from the conductivity sensor on CTD. | |
| CTDOXY | Dissolved oxygen measured from sensors on CTD. | µmol kg⁻¹ |
| OXYGEN | Dissolved oxygen measured from discrete samples. | µmol kg⁻¹ |
| TA | Total Alkalinity measured from discrete samples. | µmol kg⁻¹ |
| DIC | Dissolved Inorganic Carbon measured from discrete samples. | µmol kg⁻¹ |
| Silicate | Silicate measured from discrete samples. | µmol kg⁻¹ |
| Nitrate | Nitrate measured from discrete samples. | µmol kg⁻¹ |
| Nitrite | Nitrite measured from discrete samples. | µmol kg⁻¹ |
| Phosphate | Phosphate measured from discrete samples. | µmol kg⁻¹ |
| DELO18 | Stable oxygen isotope ratio measured from discrete samples. | ‰ |

**Table 2: Parameters included in this data product (adapted from Jiang et al., 2021).**

When samples are collected at i stations, a half value is used. For example, at station GAK1i, the sample ID is 1.5, giving the sample collected at GAK1i, cast 3, Rosette position of 11 a Sample ID of 15311. Stations visited off the GAK Line,



including repeat stations visited in most years within Prince William Sound (PWS, Fig. 1) and stations of opportunity, are not included in this data product. See Sect. 4 for additional information on stations and data collected outside the GAK Line.

Quality control (QC) steps on the cruise–level files follow previous studies (e.g., Tanhua et al., 2010, and Jiang et al., 2021). First, the 'zero step' QC is performed on individual measurements from instruments during collection. Next, the primary QC level used property–property plots to eliminate any outliers. Finally, we added an additional step to identify questionable data based on expected values using the regression according to Evans et al. (2013). Here we applied a multiple linear regression (MLR) based on an $O_2$–based algorithm to predict DIC. Then, we compared the derived DIC from our predictions

to measured DIC to isolate measurements where the difference between measured and predicted values exceeded 4 times the root mean square error (RMSE) of the algorithm. The final QC step applied in this dataset is not comparable to the secondary level QC flags for global products such as the Global Ocean Data and Analysis Product (GLODAP, Lauvset et al., 2022). Both GLODAP and CODAP–NA use an additional QC process to identify biases in the data (Olsen et al., 2016 and 2017). The data in this product have not been converted or corrected in any way for potential systematic biases.

Quality flags (QF) were applied to the cruise–level files according to best practices outlined in Jiang et al. (2022) and summarized in Table 3. This data product and corresponding figures in this publication only include GAK Line data when the measured carbonate parameters (TA and DIC) had a QF = 2 ("acceptable"). No data identified as QF = 3 ("questionable") or QF = 4 ("known bad") are included in the merged data product. Due to inconsistencies in nutrient sample collection and analysis methods on samples from cruises in 2008, 2009, 2010, and 2012, these data were assigned a QF = 3

in the original cruise–level files and are also excluded from this data product (indicated by the * in the nutrients column in Table 1). This data product (Monacci et al, 2023) used "–999" and QF = 9 ("not reported) when the cruise–level data QF $\overline{}$ 2. The individual cruise datasets include all data (Monacci et al, 2020a–j).

| Flag | Meaning |
|------|---------|
| 2 | Acceptable |
| 3 | Questionable |
| 4 | Known Bad |
| 6 | Average value |
| 9 | Missing value |

**Table 3: Summary of standard primary level quality control flags (Jiang et al., 2022) in this data product.**

**2.3 Calculations and Error**

Uncertainties in derived marine carbonate parameters are rarely reported and rely on propagating the uncertainties in both the measured parameters and the equilibrium constants. The analytical precisions for TA and DIC measurements were 1.71 ± 0.46 μmol kg⁻¹ and 1.10 ± 0.37 μmol kg⁻¹, respectively, with these precisions determined by averaging standard deviations



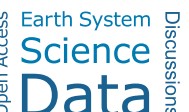

of replicate measurements of CRMs required to calculate correction factors that are applied to the instrumental raw data on a daily basis. The accuracy of TA and DIC measurements were $2.24 \pm 1.23$ μmol kg$^{-1}$ and $2.90 \pm 0.96$ μmol kg$^{-1}$, respectively.

Accuracy was determined by running CRMs as a sample bottle (CRM$_B$), following the method in the Bockmon and Dickson (2015) inter–laboratory comparison, and calculating the difference between the measured CRM$_B$ value, using the daily CRM correction factor, and the known value of the CRM$_B$. CRM$_B$ were not used in the daily correction factors and were generally run at least once per year and when any major repairs or replacements were made on the VINDTA or AIRICA. The CRM$_B$ technique was used extensively when the AIRICA was acquired and the OARC began moving away from analyzing DIC on

the VINDTA 3C paired with the Coulometer. The mean analytical uncertainty for our measured variables, TA and DIC, are $2.82$ μmol kg$^{-1}$ and $3.10$ μmol kg$^{-1}$, respectively (Table 4).

| Measured | Mean uncertainty |
|---|---|
| $f$CO$_2$ | 2.7 μatm |
| $\Omega_A$ | 0.25 |

| Calculated | Mean uncertainty |
|---|---|
| TA | 2.82 μmol kg$^{-1}$ |
| DIC | 3.10 μmol kg$^{-1}$ |

**Table 4: Mean uncertainties of parameters calculated in CO2SYS. Input parameters are *in situ* T, S, and pressure and measured TA, DIC, phosphate and silicate. Mean analytical uncertainties for the measured parameters DIC and TA are in quadrature, combining the accuracy and precision of the instruments.**

Using the analytical accuracy and precision of the Marine Analytics and Data (Marianda) built instruments, VINDTA and AIRICA, the uncertainty was calculated in quadrature using Eq. (2):

$$Uncertainty = \sqrt{accuracy^2 + precision^2} \qquad (2)$$

The mean handling error for TA and DIC are $5.54 \pm 3.15$ μmol kg$^{-1}$ and $5.54 \pm 2.74$ μmol kg$^{-1}$, respectively, which is larger than the mean analytical uncertainty. Mean handling error, calculated from the triplicate discrete samples collected from the

same Niskin bottle on most cruises (Table 1) are assumed to be representative of both sampling and storage uncertainties. Analytical precision and accuracy and the handling error we report here are averaged for all 20 cruise–level datasets.

Due to a low number of samples collected below 1500 m water depth, we were not able to perform a crossover analysis on deep water property variability. Therefore, the most recently developed tool (Jiang et al., 2021) for inter–cruise comparisons and internal consistency was not used, and we do not have an inter–cruise error estimate as produced on other multi–cruise

projects (e.g., Gouretski and Jancke, 2001; Johnson et al., 2001; Olsen et al., 2016). The internal consistency of the dataset is challenged in multiple ways, including the various methods used to sample and analyze nutrient measurements, as discussed above. However, it also routinely challenged the capacity of samplers to collect data relevant for inter–cruise comparisons.



Deep water sampling requires wire time not just for the CTD cast itself, but also weather windows and conditions that allow for sampling at the deepest station (GAK15). Accordingly, deep water data in this dataset that could have been used for the calculation of internal consistency was only available in the early part of the dataset (before 2014). Furthermore, the low sample number (n; Table 5) of these samples also challenges the creation of a statistical mean against which a potential bias of each cruise could be measured. Accordingly, no bias in individual cruises was identified, and no correction for internal consistency was applied to any single mission in this dataset. Despite these high errors in deep water sampling, note that there is sufficient surface water data to explore climatological changes with statistical significance elsewhere in this dataset.

| Season | Sample depth (m) | TA ($\mu$mol kg$^{-1}$) | DIC ($\mu$mol kg$^{-1}$) | n |
|---|---|---|---|---|
| All | < 950 | 2378 ± 21 | 2357 ± 25 | 52 |
|  | 999-1022 | 2375 ± 23 | 2355 ± 27 | 34 |
|  | >1400 | 2390 ± 12 | 2366 ± 16 | 13 |
| Spring | < 950 m | 2387 ± 29 | 2367 ± 22 | 18 |
|  | 999-1022 | 2385 ± 34 | 2367 ± 25 | 12 |
|  | >1400 | 2392 ± 10 | 2368 ± 13 | 6 |
| Autumn | < 950 | 2375 ± 17 | 2352 ± 27 | 36 |
|  | 999-1022 | 2370 ± 11 | 2349 ± 26 | 22 |
|  | >1400 | 2389 ± 14 | 2364 ± 19 | 7 |

**Table 5. Deep water data used for the demonstration of internal consistency, including the average and standard deviation of DIC ($\mu$mol kg$^{-1}$) and TA ($\mu$mol kg$^{-1}$) data below 950 m, between 999 and 1022 m, and deeper than 1400 m across the entire dataset, in Spring, and in Fall. The number of samples used to derive these statistics is also shown.**

We used CO2SYS v1.1 (van Heuven et al., 2011) to calculate the fugacity of carbon dioxide ($f$CO$_2$), pH on Total Scale (pH$_T$), and saturation state for aragonite ($\Omega_A$) when the measured input variables (T, S, DIC, TA, phosphate, and silicate) had a QF = 2. In years where no nutrient data met the QF = 2 criteria (e.g., 2008, 2009, 2010, 2012), we included a seasonal average nutrient value for that location and depth in the calculation. The nutrient alkalinity increases the $f$CO$_2$ and reduces the carbonate ion (CO$_3^-$) value when the DIC–TA input pair is used (Orr et al., 2015), therefore, the averaged nutrient values are a better estimation than not including phosphate and silicate values. We applied the K$_1$ and K$_2$ stoichiometric constants from Millero et al. (2006), the K$_{SO4}$ dissociation constant from Dickson (1990), and the K$_B$ constant from Uppström (1974). New software packages have the capacity to compute propagated uncertainties (Orr et al., 2018; Sharp and Byrne, 2021; Dillion et al., 2020) for the calculated parameters. Using the Orr et al. (2018) application with average input values from this data product and the reference uncertainties results in percent relative combined standard uncertainty of nearly 4%, for $f$CO$_2$ (Table 4).



**2.4 Determining controls on surface $f$CO$_2$**

We follow the approach by Wang et al. (2022), a modified method first developed by Takahashi et al. (2002), to calculate the interannual $f$CO$_2$ variability drivers, as well as the influence of sea surface temperature (T) and all other non–thermal processes (nT) on the $f$CO$_2$ anomaly using Eq. (3) and Eq. (4):

$$T(\Delta fCO_2) = fCO_2 - nfCO_2 \tag{3}$$

$$nT(\Delta fCO_2) = nfCO_2 - fCO_2(mean) \tag{4}$$

Like Wang et al. (2022), the T and nT in this study are different from the thermal and non–thermal $f$CO$_2$ components calculated in Takahashi et al. (2002) due to the unavailability of annual mean $f$CO$_2$ in high latitudes. Note that Eq. (4) is different from Wang et al. (2022) as we are calculating seasonal means, not ocean-atmosphere flux. Therefore, the quantity $f$CO$_{2(mean)}$ represents the climatological spring or autumn means of $f$CO$_2$ and n$f$CO$_2$ is the temperature–normalized $f$CO$_2$ relative to the climatological mean spring or autumn temperature using Eq. (5):

$$nfCO_2 = fCO_2 \times exp\,(0.0423(T_{mean} - T_{obs}) \tag{5}$$

Equation (5) is from Takahashi et al. (2002), where T$_{obs}$ is the *in–situ* temperature, and T$_{mean}$ (Fig. 4) is the climatological spring or autumn mean value. Wang et al. (2022) explains that Eq. (3) is a measure of the temperature effect, relative to the climatological seasonal mean temperature and Eq. (4) represents the impact of processes (e.g., biological CO$_2$ consumption or respiration, ocean-atmosphere CO$_2$ flux, and changes in carbonate ion concentrations) unrelated to temperature (e.g., Bates et al., 2013, Cross et al., 2013). Lastly, we calculate the $f$CO$_2$ anomaly using Eq. (6):

$$fCO_2\,anomaly = fCO_2ocean - fCO_2mean = T(\Delta fCO_2) + nT(\Delta fCO_2) \tag{6}$$

**3 Dataset Demonstration**

In this section, we offer some examples of the value and potential usages of this dataset for evaluating carbonate chemistry patterns and discuss how additional data collections may improve the quality, usability, and accessibility of the dataset. At present, we estimate that the value of this time series will be primarily for the exploration of climatological patterns in carbonate chemistry parameters. Below, we offer a climatological exploration of $f$CO$_2$ as an example. In general, we find that this dataset is of sufficient length and quality to determine some climatological trends in $f$CO$_2$, as well as to assess some of the potential physical drivers of this variability. However, these physical parameters do not sufficiently describe these emerging trends. Additional datasets and analysis will be necessary to identify the mechanisms that underpin these changes. Overall, this dataset is good for exploring climatological mean states, some geochemical trends and variability, the



Fig. 4: a) The climatological average seawater temperature in the upper 250 m of the GAK Line transect in a) spring (May) and b) autumn (September) during all study years (2008–2017), plotted with salinity contours.

physicochemical contributions to this variability, but not for assessing potential biological contributions to carbon system variability.

### 3.1 Seasonality of $f$CO$_2$

The average spring (a) and autumn (b) $f$CO$_2$ along the GAK Line are shown in Fig. 5. In general, seasonality was more evident in the intermediate depths (50–200 m), likely due to the known seasonal oscillation in downwelling strength. For example, 50% of autumn $f$CO$_2$ values fell in the 683–980 µatm range, which was slightly higher than the values from spring (600–967 µatm). By contrast, the average surface $f$CO$_2$ did not vary seasonally. The surface $f$CO$_2$ in the top 50 m in spring ranged from 247 µatm to 450 µatm, with 50% of data ranging from 309 to 363 µatm, lower than the atmospheric level. The



Fig. 5: a) The average fugacity of carbon dioxide ($fCO_2$) in the upper 250 m of the GAK Line transect in a) spring (May) and b) autumn (September) during all study years (2008–2017), plotted with salinity contours.

average surface $fCO_2$ in autumn was almost identical to spring, but with a few extreme outliers (460–528 µatm) because of episodic mixing events that brought the higher $fCO_2$ from the deep layer to the surface. The median $fCO_2$ in surface 50 m in spring and autumn were 329 µatm and 336 µatm, respectively. This surface $fCO_2$ value is lower than atmospheric $fCO2$, again indicating this region is a year–round $CO_2$ sink, as previously documented (Evans & Mathis, 2013).

## 3.2 Climatology of surface $fCO_2$

The climatological average $fCO_2$ in the surface 50 m was lower than the atmospheric $fCO_2$ value (Fig. 6), with episodic high $fCO_2$ events (>450 µatm) in both seasons. Interestingly, there were more episodic high $fCO_2$ values before 2014 in both seasons. Based on a temperature normalization analysis, temperature seems not to be the primary driver of either the long–term decline in average surface $fCO_2$ or the frequency and magnitude of the episodic high $fCO_2$ events. The T($\Delta fCO_2$)





represents the impact of temperature anomalies on surface $f$CO$_2$. As temperature anomalies increase, $f$CO$_2$ and T($\Delta f$CO$_2$)

increase first due to a decrease in CO$_2$ dissolution. This is further compounded by shifts in equilibrium that favor reactants,

specifically, CO$_2$ and H$_2$O, over their product forms (H$_2$CO$_3$, HCO$_3^-$, and CO$_3^{2-}$). Consequently, more CO$_2$ remains in its

initial, unreacted state. However, there were no clear temporal changes from T($\Delta f$CO$_2$) between before 2014 and after 2014,

anomalies were less than 50 µatm, which cannot explain the >200 µatm episodic changes. Therefore, our analyses suggest

that other nonthermal processes, such as anomalies in productivity/respiration, may well lead to interannual variations in

surface $f$CO$_2$. Anomalies in the patterns of primary productivity and respiration may, at least to a first order, be related to

patterns in regional wind forcing, and we note that 2011–2013 displayed some of the weakest CDUI values over the start of

each year in this dataset (Fig. 2). Weaker CDUI values over the first half of the year would imply reduced winter

downwelling and storm activity, which acts to replenish surface nutrients. A reduction in stormy conditions would pre-

condition the system for weaker spring phytoplankton blooms, either by weaker nutrient replenishment or a change in the

phenology of the spring phytoplankton bloom to groups that favor more stratified conditions. Such a state could lead to a

weaker and shallower extent of under-saturated surface water with respect to the atmosphere, which then may be easily

mixed by episodic wind forcing and lead to the observed $f$CO$_2$ anomalies. The relationship between the magnitude and

frequency of these high $f$CO$_2$ events, the relaxation of downwelling, and influences on primary productivity requires

additional investigation.

### 3.2 Spatiotemporal trends of CO$_2$

In addition to showing the climatological average spring and fall modes of $f$CO$_2$ variability, the time–series component of

this dataset can also be used to understand the CO$_2$ sink changes over time. In Fig. 7, we split the data into three regional

components to show emerging trends in the upper 50 m of different sub–regions of the Seward Line (Fig. 1): nearshore

(GAK 1–2), middle (GAK 3–7), and offshore (GAK 8–15). If the surface CO$_2$ increases have been slower than in the

atmosphere, this region has been a decreasing CO$_2$ sink during the studied period. Sutton et al. (2019) has reported the Time

of Emergence (TOE) of anthropogenic CO$_2$ change depends on the internal variability, which was generally >20 years in

coastal regions. Therefore, we acknowledge that the temporal span of these observations is from 2008 to 2017, which may not

be sufficient to represent the long–term anthropogenic trends. Indeed, the CO$_2$ trend was insignificant in the nearshore regions

in neither spring nor autumn because of the high interannual variability (Fig. 7). While the anthropogenic CO$_2$ trend did not

emerge from the short record in autumn or nearshore subregions, we did find significant variability in the middle and offshore

subregions: the surface CO$_2$ has been significantly decreased in the spring. As discussed with Fig. 6b, the influence of

temperature anomalies on the $f$CO$_2$ anomaly is minor compared to nT($\Delta f$CO$_2$), so the consistent T cannot t explain the

decreasing $f$CO$_2$ trend: Enhanced primary productivity should support the lower surface $f$CO$_2$ (~250 µatm) in nearshore and

offshore regions after 2014. The increasing primary productivity made this area a rising carbon sink in spring after 2014.

However, there is not sufficient information in this dataset to confirm this hypothesis. Sampling in May and September does





Fig. 6: The fugacity of carbon dioxide ($f$CO₂) is impacted by sea surface temperature (T, Eq 3) and other non–thermal processes (nT, Eq 4). Panels show the upper 50 m of the water column average along the GAK Line transect from 2008 to 2017 for the a) spring $f$CO₂, b) thermally driven spring $f$CO₂ anomaly [T (Δ $f$CO₂)], c) non–thermal spring $f$CO₂ anomaly [nT (Δ $f$CO₂)], d) autumn $f$CO₂, e) autumn T (Δ $f$CO₂), and f) autumn nT (Δ $f$CO₂).








**Fig. 7: Regional (nearshore, middle, offshore) violin plots of the upper 50 m $f$CO₂ and temperature anomaly for**
**spring (a and c) and autumn (b and d). Seward Line stations (Fig. 1) are defined in regions: nearshore (GAK1–2),**
**middle (GAK3–7), and offshore (GAK8–15).**

not sufficiently bracket the spring bloom and is too long for the reliable calculation of seasonal net community production

from DIC or nutrient measurements based on best practices (e.g., Mathis et al. 2009). Combining the data shown here with

moored datasets, such as from GAKOA (Monacci et al., 2022), and remotely sensed biological parameters, such as

chlorophyll, may be a fruitful line of future enquiry.



## 4 Data availability

The digital object identifier (DOI) for this merged data product is https://doi.org/10.25921/x9sg–9b08 (Monacci et al., 2023). Users looking to access marine carbonate parameter data for the GAK Line that have been designated as good (QF = 2),

should consider using this data product. The cruise–level DOIs listed in Table 1 are archived at the National Oceanic and Atmospheric Administration (NOAA) National Centers for Environmental Information (NCEI) (Monacci et al., 2020a–j). These data can also be accessed from the NOAA NCEI Ocean Carbon and Acidification Data System project page: Seward Line Cruises 2008–2017 www.ncei.noaa.gov/access/ocean-carbon-acidification-data-system/oceans/Coastal/seward.html. Users interested in all available data should use the cruise–level data at NCEI (Table 1). Users should also be aware that the

GAK Line was visited during the NOAA Ship *Ronald H. Brown* cruise RB1504 (EXPOCODE 33RO20150713) in July 2015. The RB1504 cruise was funded by the NOAA Ocean Acidification Program and data are archived under the DOI https://doi.org/10.25921/dey6-9h45 (Cross et al., 2019).

Data collected during 2008–2012 have been used by Shake (2011) to describe seasonal variability, Evans and Mathis (2013)

to determine the Gulf of Alaska as a $CO_2$ sink, Evans et al. (2013) to develop a regression modeling approach to understand variability, and Siedlecki et al. (2017) and Hauri et al. (2020) to validate regional models. The cruise–level data sets (Table 1) include observations from the western PWS (Fig. 1) but are not included in this merged data product for the GAK Line. Carbonate system data collected in PWS on May and September cruises from 2009 to 2012 were reported in Evans et al. (2014) and Cai et al. (2021). The datasets were not publicly archived at the time of these publications and future uses and references

should include the appropriate dataset citations (Monacci et al., 2020a–j). There continues to be a shortage of dataset citations in publications (Vannan et al., 2020), and this data product is direct proof of our support to make data accessible, usable, and citable. We ask users to invite collaboration with the scientific investigators on these datasets as they can provide additional insight. Citation and collaboration are not only good practice but will promote sharing of data and advancing accessibility.

## 5 Conclusion

This data product provides a distinct data set for users interested in the NGA ecosystem, coastal carbon dynamics, ocean acidification, biogeochemical cycling, and ocean change. The data product can be used to explore natural environmental variability on the physical system and their impacts on marine carbonate parameters. Our examples provide assessments of seasonal, interannual, and regional differences. This data product's timespan is too short to explore the rate of anthropogenic $CO_2$ absorption, though there are sustained observations in the region using a moored autonomous $pCO_2$ (MAPCO2™,

Sutton et al., 2019) system and through the National Science Foundation's Northern Gulf of Alaska Long–Term Ecological Research program (NGA–LTER). Ocean time–series allow us to understand long–term change and can be an important tool in developing predictions and solutions to continued anthropogenic perturbations. As the global economy turns to the ocean for additional $CO_2$ sequestration in marine carbon dioxide removal (mCDR) methods, our reliance on time–series sites for



our calculations are likely to expand. This data product fills a large gap in the publicly accessible inorganic carbon
measurements for a subarctic environment. While our efforts have fallen short of some best practices (e.g., discrete salinity
and nutrient measurements) outlined in GLODAPv2 (Olson et al., 2016), we feel our carbonate parameter measurements are
consistent with the OA community goals and standards.

**Author contributions**

NMM, JNC, WE, and HW contributed to writing and editing. NMM prepared the manuscript and is responsible for the data
management, archival, and updates. NMM, JNC, and WE conducted the data QC and error analysis. HW and JNC produced
the controls on $CO_2$. JTM initiated the SL OA program. JNC and JTM were PIs on grants awarded. NMM and WE
participated in field work.

**Competing Interests**

The authors declare that they have no competing interests.

**Acknowledgements**

We acknowledge and respect that the University of Alaska Fairbanks Troth Yeddha' campus is located on the current,
traditional, and ancestral homelands of the Dene people of the lower Tanana Valley and that our study area encompasses the
current, traditional, and ancestral region of the Sugpiaq and Alutiiq people. This work would not have been possible without
support from the crew of the USFWS R/V *Tiĝlax̂*. We acknowledge our co–contributors to the 10, unique datasets produced
from 20 research cruises including Seth Danielson, Russ Hopcroft, Calvin Mordy, Daniel Naber, Kristen Shake, Katherine
Trahanovsky, Thomas Weingartner, Terry Whitledge, and Eric Wisegarver. We thank the Alaska Ocean Observing System
under NOAA awards A08NOS4730406, NA11NOS0120020, and NA16NOS0120027 for financial support of the discrete
inorganic carbon analyses and the Exxon Valdez Oil Spill Trustee Council, Gulf Watch Alaska, and the North Pacific
Research Board for financial support of the research cruises. This manuscript is PMEL contribution number 5508; thank you
to Dr. Simone Alin for the internal review.

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
