# Peer review of "A decade of marine inorganic carbon chemistry observations in the northern Gulf of Alaska – Insights to an environment in transition"

_Earth System Science Data, 2023_

## Author Comment (AC1)

**General comments**

This paper introduces a dataset of physico-chemical parameters (pressure, temperature, salinity, oxygen concentration) collected by CTD, coupled with measurements in lab of oxygen concentration, Total Alkalinity, Dissolved Inorganic Carbon and selected nutrients (phosphate, nitrate, nitrite, silicate). The data are original and come from cruises that sampled biannually the same line in the Gulf of Alaska from 2008 to 2017.

All considered, this is an interesting paper that supports a solid dataset and therefore in my opinion it is publishable after a few corrections, mainly typographic, are carried out.

**Specific comments**

The manuscript

The manuscript is written in good English and the reading flows smoothly. The authors explain their work extensively, but concisely, so that the length of the paper is adequate to its content. The manuscript features 7 figures, all of them more or less useful in facilitating the comprehension of the text. As far as the form is concerned, in my opinion overall it is a very well written paper.

The introduction adequately frames Ocean Acidification progress and consequences, and the importance of datasets to comprehend the phenomenon, citing a fair number of papers as references. However, at line 61 begins a sentence that I have somehow to contest. When I read it, I understand that phytoplancton primary production concurs in favouring $\Omega_A < 1$, which doesn't sound right. Maybe the sentence should be rewritten to better clarify the thought of the authors.  Done.

Methods are well explained, with all the information needed. The organization and manipulation of the dataset are described in depth, not hiding possible fallacies of the data, and the tables included in the description are essential to the usability of the dataset. In the last two sections, equations are clearly written and explained. My only remark concerns the second analysis of silicate, which is introduced to the reader, but whose results are not shown. Clarified on Line 191.

In the following chapter, the authors demonstrate how the dataset can be used to calculate the fugacity of $CO_2$ and to study its correlation to a physico-chemical

parameter like temperature. The example is well done and is able to highlight the potentiality of the dataset.

Data availability is complete and redirects the reader to available datasets related to this one. This chapter also relates how the dataset has been used to produce scientific publications.

Conclusions and following chapters are fine.

The dataset

The dataset supported by the manuscript is the result of the merging of 20 datasets of data collected from cruises biannually from 2008 to 2017 along a line on the continental shelf of the Gulf of Alaska. In the process, data with a Quality Flag "Acceptable" have been retained and data with a Quality Flag "Questionable" or worse have been expunged.

The dataset presents an impressive amount of data, not for the number of measured variables, but rather for the number of samplings. The authors follow the best practices data standards outlined in Jiang et al. (2022). The format is universal (csv) and the dataset is easily manageable, so that I had few problems in elaborating it to suit my needs.

There however a few remarks to be done.

- The names of the stations are an alphanumeric text (e.g., GAK1). Since in alphabetic sorting rules are different from numerical sorting, the final result is that alphabetical sorting of Station_Name is impossible (GAK10 is put between GAK1 and GAK2 instead after GAK9). I suggest adding a 0 in front of the single digit (GAK01).  We agree that this is a better naming scheme, though making this change this would make it incompatible with the 50+ year records, as all other datasets do not use the "GAK01" suggested format.  We respectively decline to make these changes to keep in tradition with the larger project naming choices (https://nga.lternet.edu/about-us/site-history/) and to more easily allow other project members to do lookups using the traditional naming scheme.

- There are two stations (RES2.5 and GAK2i) with the same Station_ID (2.5). Even if Jiang et al. (2022) only ask for a numerical value and not for uniqueness, confusion may arise with the calculated Sample_ID. My suggestion is to change RES2.5 Station_Name from 2.5 to 0, if possible.  This suggestion came up during our deliberations before this product's submission for review.  The 20 individual cruise files (https://www.ncei.noaa.gov/access/ocean-carbon-acidification-datasystem/oceans/Coastal/seward.html), which are not limited to the GAK Line as this data product is, include several repeating Station ID's: GAK2, PWS2, CB2, KIP2, and HB2 all have Station ID = 2.  Since the Sample ID is still unique to each EXPOCODE, even with duplicative Station ID's, the current identifying scheme is acceptable.  We respectively choose not to rename the Station ID for RES2.5 for two reasons.  First, all 20 individual cruises files would also need to adhere to this recommendation, making the stations exampled above to be renumbered, falling outside of the intuitive station number that match maps and decadal records from this project.  The second, is that there is a Resurrection Bay site 0 (RES0; though not sampled for carbonate chemistry) which is a different site, further north of RES2.5, and would be confusing for project members familiar with the long term project.  We have added text to further clarify this.

- Date (UTC). This column is not included in Jiang et al. (2022), but it can still be useful. I find however a little disconcerting that part of the dates is formatted on the right and part on the left of the column. I understand that springs are on the right and autumns on the left, but still... Anyway, spring 2016 is on the wrong side and the name of the column might need an underscore (Date_(UTC)).  We have been unable to duplicate the circumstances where the data is aligned differently depending on year.  These files are submitted through OCADS and the downloadable file the public can access is not the exact file submitted by data authors, which is likely why we cannot replicate this experience.  This may be related to the version of software used to view the .csv file.  We have edited the Date column header to "Date_UTC" as suggested.

- Units of measurement. The only way to know the units of measurement used in the dataset is to read the paper, while in the original datasets the units were indicated in the header. I prefer when units are explicitated in the dataset, if there are not constraints against it.  Thank you for this suggestion.  We have added the units row to this merged data product, which now matches the units row for the 20 individual cruise files.

- Time_(UTC). This column is useless and could be erased. Done.

**Specific comments**

Line     Error

- 046      Reisdorph et al., 2014 should be Reisdorph and Mathis, 2014 Done.

113     Bakun 1973, 1975 should be Bakun, 1973 and 1975 Done.

135     Jacox, 2018 should be Jacox et al., 2018 Done.

236     typo: when the cruise–level data QF -2 Done.

289     Dillion should be Dillon Done.

369     I am not a native English speaker, but to me the sentence "trend was insignificant in the nearshore regions in neither spring nor autumn" sounds better "trend was insignificant in the nearshore regions in both spring and autumn" Done.

373     typo: so the consistent T cannot t explain Done.

374     Enhanced should be enhanced Done.

448     CO2 should be $CO_2$ Done.

454     missing italic Done.

461     missing italic (x2) Done. Done.

461     missing italic Done.

484     missing italic Done.

508     CO2 should be $CO_2$ Done.

520     CO2 should be $CO_2$ Done.

554     CO2 should be $CO_2$ Done.

558     missing italic Done.

594     missing italic Done.

611     CO2 should be $CO_2$ Done.

624     CO2 should be $CO_2$ Done.

712     CO2 should be $CO_2$ Done.

734     CO2 should be $CO_2$ Done.

740     CO2 should be $CO_2$ (x2) Done.

747     CO2 should be $CO_2$ Done.

752     CO2 should be $CO_2$ Done.

764     missing italic Done.

Reply

**Citation**: https://doi.org/10.5194/essd-2023-325-RC1

---

## Author Comment (AC2)

**RC2**: 'Comment on essd-2023-325', Anonymous Referee #2, 06 Nov 2023

This paper offers a valuable dataset encompassing decadal marine inorganic carbon chemistry observations in the northern Gulf of Alaska (NGA). The NGA, with its diverse ecosystem, including significant commercial fisheries, serves as a crucial intersection for discussions on ocean acidification (OA) processes in high-latitude coastal waters. The assembled decadal time-series product presented in this study provides a resource for researchers aiming to comprehend OA under climate change, assess the drivers of coastal OA, and evaluate biogeochemical performance. The paper meticulously describes the dataset, covering aspects of data collection, quality control, and uncertainty estimations. While the overall presentation is robust, I have a few specific comments regarding the manuscript:

Major Comment:

why pH (even calculated values) is not included in this dataset.

We chose to include only measured variables in the data product. Users of the data product can choose to use CO2SYS, as we have demonstrated, or other packages, which include various input choices depending on the version history. Regarding input choices, work on dissociation constants is constantly being updated (e.g. Waters et al., 2014) and certain input variables are more appropriate for coastal data influenced by freshwater (e.g. Millero et al., 2010).

Minor Comments:

1. Clarify $\Omega_A$ and $\Omega_C$, different forms of calcium carbonate mineral saturation state in Line 37, following the definition of $\Omega$, given their use in subsequent contexts. Done.

2. In Table 4, $fCO_2$ and $\Omega_A$ should be represented as calculated parameters, while TA and DIC should be indicated as measured variables. Please adjust the variable names in the first column accordingly. Done.

3. Please provide an explanation of how mean uncertainties in Table 4 were calculated (also mentioned in Line 251). If these uncertainties are derived from Equation (2), it would be helpful to present this equation earlier. Additionally, include a sentence explaining the mean uncertainty of calculated parameters. Done.

4. Section 2.4 of the Methods is not entirely clear. Specify whether $fCO_2$ refers to sea surface $fCO_2$. It is confusing to add "not ocean-atmosphere flux" in Line 300 because the flux is typically represented by $FCO_2$. In addition, is $fCO_2$ the same as

fCO$_2$ (ocean). Also, the subscripts of this session should be consistent. Some are subscripts (like T), and some are like annotations with/without "()".  Done.

5. Line 317, What physical parameters? Done.

6. Lines 330-337: Specify whether the 'top 50m' refers to the mixed layer depth. If this depth pertains to the mixed layer, clarify this term, as '50m' appears multiple times. When referring to surface water, it would be helpful to specify that it suggests the water mass above the mixed layer depth. Done.

7. Line 393: The DOI link provided does work.

**Citation**: https://doi.org/10.5194/essd-2023-325-RC2